# Show, Write, and Retrieve: Entity-aware Article Generation and Retrieval

**Zhongping Zhang**     **Yiwen Gu**     **Bryan A. Plummer**
Boston University
{zpzhang, yiweng, bplum}@bu.edu

## Abstract

Article comprehension is an important challenge in natural language processing with many applications such as article generation or image-to-article retrieval. Prior work typically encodes all tokens in articles uniformly using pre-trained language models. However, in many applications, such as understanding news stories, these articles are based on real-world events and may reference many named entities that are difficult to accurately recognize and predict by language models. To address this challenge, we propose an ENtity-aware article GeneratIoN and rEtrieval (ENGINE) framework, to explicitly incorporate named entities into language models. ENGINE has two main components: a named-entity extraction module to extract named entities from both metadata and embedded images associated with articles, and an entity-aware mechanism that enhances the model's ability to recognize and predict entity names. We conducted experiments on three public datasets: GoodNews, VisualNews, and WikiText, where our results demonstrate that our model can boost both article generation and article retrieval performance, with a 4-5 perplexity improvement in article generation and a 3-4% boost in recall@1 in article retrieval. We release our implementation at this http URL.

## 1 Introduction

Comprehending articles enables a wide range of applications such as story generation (Fan et al., 2018; Peng et al., 2018), image-to-text retrieval (Tan et al., 2022), automated journalism (Leppänen et al., 2017; Brown et al., 2020), defending against misinformation (Zellers et al., 2020; Tan et al., 2020), and writing Wiki articles (Banerjee and Mitra, 2016; Stephen et al., 2017), among others. Inspired by the impressive capability of large language models, recent work (*e.g.*, Radford et al. (2019); Brown et al. (2020); Wang and Komatsuzaki (2021)) generate or retrieve articles by training language models on massive datasets (*e.g.*, The

Figure 1: We propose entity-aware language modeling for article generation and retrieval. Prior work (Brown et al., 2020; Zellers et al., 2020; Ouyang et al., 2022), shown in (A), typically comprehends articles by uniformly encoding all text tokens. However, these approaches face challenges in accurately recognizing and predicting named entities. In our paper, shown in (B), we propose a method to extract named entities from embedded images and explicitly model the named entities in articles, boosting the performance of both article generation and article retrieval.

Pile (Gao et al., 2020) or LAION400M (Schuhmann et al., 2021)). These models typically uniformly encode all text tokens in the articles including named entities (Radford et al., 2018; Brown et al., 2020; Zellers et al., 2020). In other words, named entities like organizations, places, and dates are modeled together with other text, as illustrated in Figure 1(A). However, it poses a challenge for these models to accurately recognize and predict named entities, as they can be unique to specific articles. *E.g.*, in Figure 1(A), entities like "Ms. Jolie" and "Jolie Pitt" may only appear in articles related to the celebrity "Angelina Jolie."

Directly extracting named entities from user-provided prompts is a straightforward entity-aware approach used by methods addressing news image captioning (Biten et al., 2019; Tran et al., 2020; Liu et al., 2021). However, these methods do not

generalize well to article generation and retrieval since they rely on substantial contextual information (via the articles) as well as a direct indication of entities that may appear in predicted captions. For example, in Figure 1 (A), the article mentions that Angelina Jolie Pitt is an Oscar winner, but this information is not present in the other metadata like the image captions. Thus, a language model generating an article must infer this named entity information rather than directly extracting it from the metadata. Even if a list of named entities were provided, an article generation model must determine where and when to use each of them. In contrast, in news image captioning, entities used in the caption almost always appear in the body of the article (Liu et al., 2021; Tan et al., 2020), and the image itself will directly inform what named entities should be used for its caption. Thus, as we will show, adapting entity-aware mechanisms from related work (*e.g.*, (Liu et al., 2021; Dong et al., 2021)) results in poor performance in our task.

To address the aforementioned issues, we propose an **EN**tity-aware article **G**enerat**I**o**N** and r**E**trieval (ENGINE) framework to explicitly incorporate and model named entities in articles. EN-GINE mainly consists two modules: a named-entity extraction module and an entity-aware module. In the named-entity extraction module, we show that providing a list of named entities for article generation improves performance. However, creating such lists does require a small overhead cost. Thus, we also demonstrate we can improve performance without manual input.

As shown in Figure 1 (B), we observe that images associated with an article often contain information about the article's events. Therefore, we explore leveraging large vision-language models to extract named entities from embedded images. Specifically, we employ CLIP (Radford et al., 2021) to automatically select a set of likely named entities from embedded images. In the entity-aware module, we introduce special tokens after each entity name to indicate its entity category. In this case, ENGINE models the named entity and its entity category jointly. An additional benefit brought by our entity-aware mechanism is the named-entity recognition (NER) ability, *i.e.*, our model not only recognizes and predicts the entity names but also predicts their entity categories simultaneously.

In summary, the contributions of this paper are:

- We propose an entity-aware language model, EN-GINE, for article generation and retrieval. Compared to existing language models (Brown et al., 2020; Zellers et al., 2020; Radford et al., 2021; Sun et al., 2023), our entity-aware mechanism enhances the recognition and prediction of named entities by jointly modeling entity names and their entity categories, boosting the performance of article generation and retrieval.
- We introduce a named-entity extraction method to recognize named entities from embedded images in articles, eliminating the overhead in manually creating a list of named entities that will appear in the articles.
- Experiments on GoodNews (Biten et al., 2019) and VisualNews (Liu et al., 2021) show a perplexity gain of 4-5 points for article generation and a Recall@1 boost of 3-4% for article retrieval. We also show that ENGINE generalizes via zero-shot transfer to WikiText (Stephen et al., 2017).
- We perform comprehensive experiments on human evaluation and machine discrimination, validating that ENGINE produces more realistic articles compared to prior work (Radford et al., 2019; Zellers et al., 2020). This suggests that our model can potentially contribute additional training data for the development of more powerful machine-generated text detectors.

## 2 Related Work

**Article Generation** in recent work uses large-scale pretrained transformer models that can be separated into unconditional text generation (Radford et al., 2018, 2019) and conditional text generation (Brown et al., 2020; Zellers et al., 2020). Generating articles via unconditional samples has been found to be less effective, since the models may interpret the first sentence of articles as a tweet and start posting responses (Brown et al., 2020). To enable controllable generation, GPT3 (Brown et al., 2020) conditions article generation on titles and the initial sentences of articles. Grover (Zellers et al., 2020) decomposes news articles into distinct parts and conditions generation on metadata like the author or organization. In this paper, we further explore the impact of named entities and embedded article images. Specifically, ENGINE produces articles conditioned on both metadata and embedded article images, with a dedicated focus on the explicit extraction and modeling of named entities.

**Article Retrieval** is commonly accomplished by image-text matching frameworks. Early work on

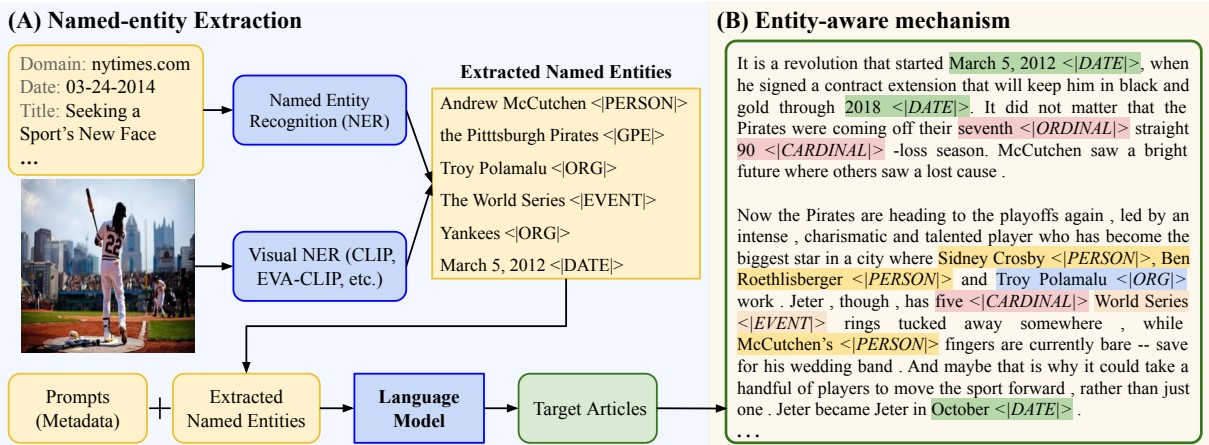

**(A) Named-entity Extraction**

Domain: nytimes.com
Date: 03-24-2014
Title: Seeking a Sport's New Face
...

Named Entity Recognition (NER)

Visual NER (CLIP, EVA-CLIP, etc.)

**Extracted Named Entities**

Andrew McCutchen <|PERSON|>
the Pitttsburgh Pirates <|GPE|>
Troy Polamalu <|ORG|>
The World Series <|EVENT|>
Yankees <|ORG|>
March 5, 2012 <|DATE|>

Prompts (Metadata) + Extracted Named Entities → **Language Model** → Target Articles

**(B) Entity-aware mechanism**

It is a revolution that started March 5, 2012 <|*DATE*|>, when he signed a contract extension that will keep him in black and gold through 2018 <|*DATE*|>. It did not matter that the Pirates were coming off their seventh <|*ORDINAL*|> straight 90 <|*CARDINAL*|> -loss season. McCutchen saw a bright future where others saw a lost cause.

Now the Pirates are heading to the playoffs again, led by an intense, charismatic and talented player who has become the biggest star in a city where Sidney Crosby <|*PERSON*|>, Ben Roethlisberger <|*PERSON*|> and Troy Polamalu <|*ORG*|> work. Jeter, though, has five <|*CARDINAL*|> World Series <|*EVENT*|> rings tucked away somewhere, while McCutchen's <|*PERSON*|> fingers are currently bare -- save for his wedding band. And maybe that is why it could take a handful of players to move the sport forward, rather than just one. Jeter became Jeter in October <|*DATE*|>. ...

Figure 2: **ENGINE overview**. Our model mainly consists of two modules: (A) Named-entity Extraction: We extract named entities from both metadata and embedded images in articles. See Section 3.2 for detailed discussion; (B) Entity-aware mechanism: Each named entity is associated by its corresponding entity category. ENGINE models the entity name and category jointly to avoid inconsistency between named entities in images and article text. See Section 3.1 for detailed discussion.

image-text matching has primarily focused on developing bespoke models (Wang et al., 2017; Gu et al., 2018; Nguyen and Okatani, 2018; Nam et al., 2017) with various retrieval loss functions, such as triplet loss (Schroff et al., 2015) or proxy anchor loss (Kim et al., 2020). However, these models are often domain-specific and limited in the expressiveness of the text. In recent work, large vision-language models (Radford et al., 2021; Sun et al., 2023; Li et al., 2022) have addressed these limitations by contrastive image-language pretraining on massive datasets (*e.g.*, LAION-400M (Schuhmann et al., 2021)). These large vision-language models have demonstrated exceptional performance in zero-shot image-text matching. Therefore, in our experiments, we employ the pretrained vision-language model EVA-CLIP (Sun et al., 2023) to extract visual and text representations from articles, and compute the cosine similarity between these representations to obtain the retrieval predictions.

**Entity-aware mechanisms** have been applied in closely related tasks, such as news image captioning, which aims to caption images based on articles and images. Ramisa et al. (2017) proposed an end-to-end framework that takes the concatenation of article and image features as input and outputs captions by an LSTM decoder. However, this approach often fails to predict named entities that were not seen during training. Thus, more recent work boosts performance by extracting entity representations from user-provided articles and inserts them into generated templates (Biten et al., 2019;

Tran et al., 2020; Liu et al., 2021). In this paper, we effectively reverse the inputs and outputs of these approaches, *i.e.*, we generate an article based on images and captions rather than generating captions based on images and articles. As discussed in the Introduction, this shift breaks the assumptions used by entity-aware mechanisms in image captioning, causing them to not generalize well in our article generation and retrieval task.

## 3 ENGINE: ENtity-aware article GeneratIoN and rEtrieval

Given user-provided prompts and embedded article images, our task aims to more accurately represent articles by recognizing and predicting named entities. Thus, we incorporate candidate named entities as an additional control to our language model. We propose an entity-aware mechanism to jointly model entity names and their corresponding entity categories in Section 3.1. To enrich the entity information available to language models, we introduce our named-entity extraction approach in Section 3.2. Finally, we introduce the learning strategy of ENGINE in Section 3.3. Figure 2 provides an overview of our method.

### 3.1 Entity-aware Mechanism

As discussed in the Introduction, accurately recognizing and predicting named entities can help avoid inconsistencies between the associated images and the textual content of an article. For example, in NBA news, an entity-aware model should be

able to predict "Curry" given the preceding word "Stephen," whereas traditional language models might struggle in this regard. Existing methods typically model named entities uniformly with other text, making the leverage of named entities less effective. To help our language model be aware of named entities, we insert the entity category predicted by SpaCy (Honnibal and Montani, 2017) after each entity name. We use special tokens as the indicator of these entity types. Then the entity name and its corresponding category are modeled jointly by ENGINE. We visualize our entity-aware mechanism in Figure 2 (B).

## 3.2 Named-entity Extraction

Our entity-aware mechanism in Section 3.1 enhances a language model's ability to recognize and predict named entities. However, we find that the entity information extracted from the metadata may not be sufficient for article generation. Thus, as shown in Figure 2 (A), we explore named entity extraction methods from various sources within articles, such as embedded images. Below we discuss two ways to create the named entity list.

**Oracle named-entities.** This approach assumes that we are provided with all the named entities that would appear in articles, *e.g.*, named entities provided by a user. To simulate user-provided named entities, we extract named entities from news articles using SpaCy (Honnibal and Montani, 2017). This list is then provided as input to our model.

**CLIP-based NER.** Existing Named Entity Recognition (NER) methods (Yadav and Bethard, 2018; Li et al., 2020) primarily distinguish named entities within text documents and are not designed for NER involving images (referred to as Visual-NER in Figure 2(A)). However, we note that CLIP (Radford et al., 2021) was trained on 400 million image-text pairs collected from the internet, many of which likely contain named entities. Thus, we use CLIP to build an open-ended Visual-NER framework. First, we construct a candidate list of named entities for each image by extracting entities from the articles in the dataset using SpaCy (Honnibal and Montani, 2017). Subsequently, we use CLIP to predict the similarity between the article images and the candidate entities. The top $k$[1] entities are then provided as input to our model.

---

[1]We set $k$ to 10 in this paper.

## 3.3 Learning Strategy

**Language Modeling.** Given a set of documents $\{x_1, x_2, ..., x_n\}$ each with variable length sequences of symbols $\{s_1, s_2, ..., s_m\}$, the statistical language model of a text document $x$ can be represented by the probability of next symbol given all the previous ones (Bengio et al., 2003):

$$p(x) = \prod_{i=1}^{m} p(s_i|s_1, ..., s_{i-1}), \qquad (1)$$

where each symbol $s_i$ is processed uniformly and the document $x$ is viewed as an unstructured *text* field (also referred as body field later). Language models based only on Eq. 1 produce articles via unconditional samples. Thus, these models are not suitable for controllable generation (Hu et al., 2017). Instead, the language model can be formulated by the joint distribution of separate fields decomposed from the article $x$ (Zellers et al., 2020):

$$p(x) = p(\text{meta}, \text{body}), \qquad (2)$$

where meta is a data-dependent term consisting of a set of subfields. For instance, meta includes *date*, *title*, *summary* in GoodNews (Biten et al., 2019) and *domain*, *date*, *topic*, *title* in Visual-News (Liu et al., 2021). Thus, we model $x$ by:

$$p(x) = p(\text{body} \mid \text{meta})p(\text{meta}). \qquad (3)$$

Based on Eq. 3, we further introduce special tokens $<$start-$\tau>$ and $<$end-$\tau>$ to indicate the boundaries of field $\tau$. The content of a target field $\tau$ is sampled from the model starting with $<$start-$\tau>$ and ending with $<$end-$\tau>$. Given the named entities extracted by our method (from Section 3.2), Eq. 3 is re-formulated as:

$$p(x) = p(\text{body} \mid \text{meta}, \text{entity})p(\text{meta}, \text{entity}). \qquad (4)$$

To sample from Eq. 4, we define a canonical order[2] among the fields (or subfields) of articles $\mathcal{F}$ : ($f_1 < f_2 < ... < f_{|\mathcal{F}|}$) and model the articles left-to-right in the order using Eq.1: $s_1^{f_1}, s_2^{f_2}, ..., s_{|f_{|\mathcal{F}|}|}^{f_{|\mathcal{F}|}}$.

**Architecture.** Following (Zellers et al., 2020), EN-GINE uses the GPT2 architecture (Radford et al.,

---

[2]We define canonical order in Goodnews (Biten et al., 2019) as: domain, date, named-entity, title, caption, summary, body; and Visualnews (Liu et al., 2021) as: domain, date, topic, named-entity, title, caption, body.

2019) for article generation. We experiment with three model sizes: (1) ENGINE-Base has 12 layers and 124 million parameters, on par with GPT2-124M and GROVER-Base; (2) ENGINE-Medium has 24 layers and 355 million parameters, on par with GPT2-355M and GROVER-Large; (3) ENGINE-XL has 48 layers and 1.5 billion parameters, on par with GPT2-1.5B and GROVER-Mega. In addition, we also show ENGINE generalizes across architectures by evaluating on LLAMA (Touvron et al., 2023). For article retrieval, we implement our method on pretrained vision-language model EVA-CLIP (Sun et al., 2023).

## 4 Experiments

### 4.1 Datasets and Experiment Settings

**Datasets.** We evaluate ENGINE on three public datasets: GoodNews (Biten et al., 2019), Visual-News (Liu et al., 2021), and WikiText (Stephen et al., 2017). GoodNews provides the URLs from New York Times from 2010 to 2018. After filtering out broken links or non-English articles, we downloaded 307,286 news articles. Following the split ratios of Biten et al. (2019), we randomly split 15,365 articles for validation, 30,728 articles for testing, and used the rest for training. VisualNews contains news articles from four news sources: *Guardian*, *BBC*, *USA Today*, and *Washington Post*. We obtain 582,194 news articles in total after we removed broken links and articles without metadata. Similarly, we get a 491,796 training set, 28,932 validation set, and a 57,889 test set. WikiText contains 600/60/60 Wikipedia articles in train/test/validation sets, respectively. We performed zero-shot article generation on the test set of WikiText.

**Metrics.** Following Zellers et al. (2020), we adopt Perplexity (PPL) [3] to evaluate models on article generation. Perplexity is defined as the exponentiated average negative log-likelihood of a sequence. Given Eq. 1, the perplexity of $x$ is calculated by:

$$\text{PPL}(x) = \exp\left\{ -\frac{1}{m} \sum_{i=1}^{m} \log p(s_i | s_1, ..., s_{i-1}) \right\}$$
(5)

where $s_1, ..., s_i$ are ground truth tokens in $x$ and $p(\cdot)$ is the probability predicted by the model. We evaluate models using Recall@K (R@1, R@5, R@10) for article retrieval.

---

[3] During inference, we get rid of entity categories from our generated articles to make fair comparisons to other baselines.

**Implementation Details** We primarily implemented our models using Pytorch (Paszke et al., 2019) and Transformer (Wolf et al., 2020) libraries. The maximum sequence length of language models is set to 1024. For ENGINE-Base and ENGINE-Medium, we used a batch size of 8 and a maximum learning rate of $1 \times 10^{-4}$. For ENGINE-XL, we used a batch size of 4 to fit into GPU memory. Correspondingly, the maximum learning rate is set to $2^{0.5} \times 10^{-4}$. We finetuned our models for around 3 epochs with 0.06 epoch for linear warm-up on both datasets. We parallelized ENGINE-XL on 4 NVIDIA RTX-A6000s and ENGINE-Medium on 2 NVIDIA RTX-A6000s. ENGINE-XL on Visual-News requires the longest training time- approximately two weeks on our system.

### 4.2 Article Generation

**Perplexity.** Table 1 presents sizes, architectures, and perplexity results of different models on Good-News (Biten et al., 2019) and VisualNews (Liu et al., 2021). We see that ENGINE variants of all model sizes significantly outperform the baselines. On the base size, ENGINE-Base(NE) improves PPL over the original GPT2-124M model by a factor of 2 (23.6→12.0, 27.5 → 13.1). We draw three major conclusions from Table 1. First, the data distribution still plays an important role. Finetuned GPT2s improve PPL over the original GPT2s. The improvements become less obvious with a greater model size (VisualNews: 27.5→18.3 of base size; 15.7→12.4 of XL size). Second, EN-GINE noticeably improves the performance over finetuned GPTs (4-5 perplexity points on both datasets), which demonstrates the effectiveness of our approach. Third, our contributions are architecture agnostic. *E.g.*, ENGINE-LLAMA outperforms LLAMA-7B with an approximately 2-point perplexity improvement on both datasets.

**Parameter Efficiency.** Table 1 shows that EN-GINE can achieve a comparable performance with alternative models using much fewer parameters. For example, ENGINE-Base(NE), with only 124M parameters, outperforms the GPT-NEO-2.7B and achieves comparable performance with finetuned GPT2-1.5B (12.0 vs. 12.6 PPL on GoodNews, 13.1 vs. 12.4 PPL on VisualNews). ENGINE-Medium (NE) model, with 355M parameters, outperforms all the GPT-Series baselines including GPT-J-6B. Figure 3 plots the perplexity as a function of the number of parameters on news datasets, demon-

| Model Name | $n_{params}$ | $n_{layers}$ | $d_{model}$ | $n_{heads}$ | GoodNews PPL ↓ | VisualNews PPL ↓ |
|---|---|---|---|---|---|---|
| GPT2-124M (Radford et al., 2019) | 124M | 12 | 768 | 12 | 23.6 | 27.5 |
| GROVER-Base (Zellers et al., 2020) | 124M | 12 | 768 | 12 | 23.8 | 21.9 |
| GPT-Neo-125M (Gao et al., 2020) | 125M | 12 | 768 | 12 | 27.1 | 29.3 |
| GPT2-124M (Finetuned) | 124M | 12 | 768 | 12 | 17.3 | 18.3 |
| ENGINE-Base (ClipNE) | 124M | 12 | 768 | 12 | 14.8 | 16.1 |
| ENGINE-Base (NE) | 124M | 12 | 768 | 12 | **12.0** | **13.1** |
| GPT2-355M (Radford et al., 2019) | 355M | 24 | 1024 | 16 | 17.8 | 20.1 |
| GROVER-Large (Zellers et al., 2020) | 355M | 24 | 1024 | 16 | 18.5 | 16.4 |
| GPT-Neo-1.3B (Gao et al., 2020) | 1.3B | 24 | 2048 | 16 | 15.3 | 15.9 |
| GPT2-355M (Finetuned) | 355M | 24 | 1024 | 16 | 13.5 | 14.0 |
| ENGINE-Medium(ClipNE) | 355M | 24 | 1024 | 16 | 11.6 | 12.5 |
| ENGINE-Medium(NE) | 355M | 24 | 1024 | 16 | **9.5** | **10.2** |
| GPT2-1.5B (Radford et al., 2019) | 1.5B | 48 | 1600 | 25 | 13.9 | 15.7 |
| GROVER-Mega (Zellers et al., 2020) | 1.5B | 48 | 1600 | 25 | 14.5 | 12.6 |
| GPT-Neo-2.7B (Gao et al., 2020) | 2.7B | 32 | 2560 | 20 | 13.5 | 14.0 |
| GPT-J-6B (Wang and Komatsuzaki, 2021) | 6B | 28 | 4096 | 16 | 11.3 | 11.6 |
| GPT2-1.5B (Finetuned) | 1.5B | 48 | 1600 | 25 | 12.6 | 12.4 |
| ENGINE-XL(ClipNE) | 1.5B | 48 | 1600 | 25 | 10.8 | 11.1 |
| ENGINE-XL(NE) | 1.5B | 48 | 1600 | 25 | **8.7** | **9.0** |
| LLAMA-7B (Touvron et al., 2023) | 7B | 32 | 4096 | 32 | 8.3 | 8.5 |
| ENGINE+LLAMA (NE) | 7B | 32 | 4096 | 32 | **6.5** | **6.4** |

Table 1: **Perplexity (PPL) comparison on GoodNews and VisualNews.** ClipNE denotes that we select CLIP-based named entities in *named-entity* field (described in Section 3.2), NE denotes that we apply oracle named entities in *named-entity* field. PPL is calculated only on the article body . We observe that our model consistently outperforms baselines of comparable model sizes.

strating that ENGINE gets as good or better results than prior work while also using fewer parameters. **Additional Baselines adapted from close-related tasks[4].** In Table 2(A), we see that ENGINE-1.5B outperforms integrating BU (Anderson et al., 2018) and VisualGLM-6B (Du et al., 2022) features, demonstrating that our entity-aware mechanism is more effective than directly incorporating image features. As discussed in the Introduction, this is likely due to the loose correlation between images and their corresponding articles. Table 2(B) demonstrates that the entity-aware mechanisms proposed for other text generation tasks do not generalize well to article generation, where our approach obtains a 4-5 PPL improvement on both datasets. We also find that InfoSurgeon struggles to generate articles well, which we argue is due to its sequence-to-

sequence translation framework finding it challenging to effectively leverage prior knowledge from pretrained language models.

**Ablation Study.** Figure 4 shows ablations of ENGINE-Base. We observe that both the *caption* and *named-entity* fields boost performance, revealing that cues from embedded images help produce higher-quality articles. Comparing using only captions (Cap) vs. combining them with our Entity-aware mechanism, we get a minimum gain of 0.6 PPL, demonstrating its effectiveness. In addition, we observe that ClipNE outperforms CapNE, validating that CLIP-detected named entities are more effective than those extracted from captions.

**Article Quality User Study.** Following (Zellers et al., 2020; Kreps et al., 2020; Brown et al., 2020), we ask annotators to distinguish machine-generated articles from human-written articles. We randomly selected 50 news stories from GoodNews and VisualNews test sets (100 total). Given the metadata and news images, we generated news articles using three different language models: GROVER-

---

[4]InfoSurgeon (Fung et al., 2021), InjType (Dong et al., 2021), VNC (Liu et al., 2021) are adapted from sequence-to-sequence translation, close-ended paragraph generation, and news image captioning, respectively. For a fair comparison, GPT2-Base is applied as the backbone of InjType, VNC, and BU for article generation.

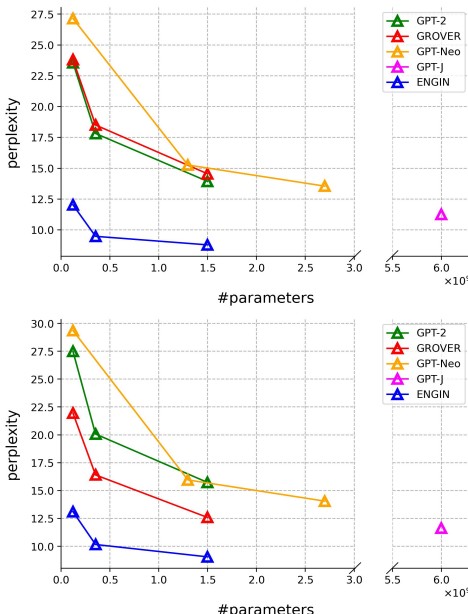

Figure 3: Comparison of the perplexity of different language models on GoodNews (top) and VisualNews (bottom) as a function of learned parameters.

| Model Name | Good-News | Visual-News |
|---|---|---|
| **(A)** BU (Anderson et al., 2018) | 26.6 | 24.3 |
| VisualGLM(Du et al., 2022) | 11.8 | 12.2 |
| ENGINE-1.5B (ours) | **8.7** | **9.0** |
| **(B)** InfoSurg (Fung et al., 2021) | 41.8 | 42.1 |
| InjType (Dong et al., 2021) | 18.2 | 19.0 |
| VNC (Liu et al., 2021) | 16.7 | 17.8 |
| ENGINE-Base (ours) | **12.0** | **13.1** |

Table 2: **PPL of baselines adapted from close-related tasks.** (A) leverages visual features directly extracted from images, (B) contrasts entity-aware mechanisms. Adapting entity-aware mechanisms from related tasks may result in poor performance in our task.

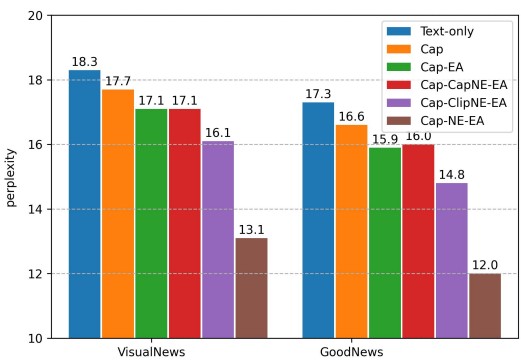

Figure 4: **Ablation results of ENGINE-Base.** Text-only denotes the model focuses only on text information, which is same to finetuned GPT2 model. Cap denotes *caption* field, EA denotes the Entity-aware mechanism. CapNE denotes named entities extracted from captions. Both the named-entity extraction module and our entity-aware mechanism can improve the performance.

Mega, GPT2-1.5B (finetuned), and ENGINE-XL. This results in a total of 200 articles per dataset. We recruited 200 Qualified Amazon Mechanical Turk (AMT) workers per dataset. Each article was annotated 5 times by AMT workers, where each worker was presented with the article titles, images and captions, and was asked to indicate if the article was human or machine generated. If they thought the article was machine-generated, they were asked to indicate a reason for it following the same option format as Tan et al. (2020). Table 3 reports annotation accuracy in identifying articles from VisualNews as machine or human-generated. We see ENGINE-XL is able to generate hard-to-detect news articles (a 2% boost over GPT2-1.5B, and a 5% gain over GROVER-Mega), validating the effectiveness of our approach.

**Machine Discriminator.** We apply OpenAI's RoBERTa (Liu et al., 2019) detector to detect generated articles. The maximum article length is cut to 512 to fit the model input size. For comparison, we use the same article set from our user study. In Table 3, we observe that the machine discriminator is much better at identifying the machine-generated news. We see that articles produced by ENGINE-XL can be reliably detected by RoBERTa though it gets the lowest accuracy on human evaluation. This can be due to the fact that GROVER-Mega, GPT2-1.5B, and ENGINE-XL all share a similar underlying model architecture. Thus, they may contain

enough similarities in the distributional features that are recognized by the machine discriminator.

**Zero-shot Article Generation** We perform zero-shot experiments on Wikipedia articles to demonstrate ENGINE'S ability to generalize. Table 4 reports performance on WikiText. We find that our oracle named entities and entity-aware mechanism still can improve the performance over several baselines, even though the data distribution between Wikipedia and news is significantly different. For example, the GPT2 models finetuned on Visualnews get worse performance than the original GPT2 models on WikiText. However, our ENGINE models get comparable or better results than the original GPT2 across different model sizes.

**Qualitative Results.** We provide a qualitative comparison of GoodNews articles in Figure 5. Con-

| Title: A Designer Unites the Basque Country |
| Domain: www.nytimes.com |
| Date: 06-10-2016 |

Jean Louis Iratzoki checking the 3D scale model of a chair for Alki.

**Human-written News Article**

In 2007, the French industrial designer Jean Louis Iratzoki opened a remote studio in a wood cabin in the foothills of the Pyrenees. His minimalist "refuge" has recently become an unlikely creative epicenter for industrial production on both the French and Spanish sides of the Pays Basque.

At last year's Maison & Objet design fair, Mr. Iratzoki won plaudits for designing one of the world's first bioplastic chairs made of a biodegradable plant-based polymer. That shell chair, part of a long collaboration with the Basque furniture brand Alki, is only one example of Mr. Iratzoki's determination to promote his native region. (Mr. Iratzoki was born in France in St. Jean-de-Luz.) He has also recently revamped the brand identity of Cazenave (a Basque chocolate company) and helped design the interiors of a luxurious eco lodge in Saubion, north of Biarritz.

This year, he founded a new practice with a Basque designer of Spanish extraction, Ander Lizaso, with the aim of creating a multipurpose design studio inspired by

**ENGINE-generated News Article**

Following in the footsteps of his male counterpart, the 2007 French - Spanish architect Jean Louis Iratzoki, whose archives became the subject of last year's Basque / Francophile Rodeo Drive exhibition, Ander Lizaso, the designer of the Pays Basque label Maison & Objet, has taken on the relatively new mantle of directoire in his own right. This year, he announced the inaugural of his own Design Angeles - in - Motion, the largest design fair for brands of the Basque / Francophone world.

" He's the one name you know everywhere, " said Lizaso, who came of age as a designer in the city of Alki during the first boom in construction trade there, selling commissioners as part of the community's growing business. " And it was the same in France. "

Going back to Belgium, where the heritage of the Maison & Objet founders still lies, imbuing his name with its celebrated high - culture associations, he explained, the house was fully aware of its Basque roots. "

**GPT2-generated News Article**

Tears eluded Chris Burch and his wife, Susana, when they first set eyes on each other as headwaiter in 1991. They had no idea how they would end up there. But neither did he, not at the time, not even 10 years later. Their friendship would grow into a tentative romance; twice, both of them, over the next 13 years, would fly away from their native Basque region to live and work and play together in New York. Until suddenly they were happily on their own, taking care of themselves. In 2012, with both 40, Burch decided to marry them. Thirty years later, the brothers will go their separate ways: Chris, the artistic director of the Gypsy Rose Lee boutique in the West Village, and Susana, founder and creative director of her own company, Flora Flora, which specializes in floral textiles and ties. "A wedding is an intimate, temporary thing, where people are sending you on your merry way," Chris explains. "You leave with a future and a few words of what you can say. With something you want to do full-fledged -- 'We're engaged' or 'I have a proposal' -- you can't even focus." The couple recently moved to Miami, where Chris wants to take a nautical theme to Hawaii, where Susana is from. Below, they discuss the undercurrents of their romance as lifelong friends.

**GROVER-generated News Article**

In both making clothing and politics, Ms. Madrid strives to combine the traditions of dress and activism with the street's distinct urbanity. She designs a range of products, from T-shirts and tees for millennials, to men's shirts for families. Everything in both stands as her identity as a designer. "I'm the most passionate person in designing my own clothes because I'm a feminist," Ms. Madrid said in an interview at a café in town. "I'm very anti-strong men and fat women."

The 40-year-old mother of two is creating a new business. It's aiming to combine traditional fashion and activism. She's hoping to raise money and offer services for political activism in the Basque country - and beyond.

In the past four decades, Ms. Madrid has been involved in social and political activism. She went on hunger strike after she was attacked while in jail. Her goal is to become a national emblem for the Basque country, she said, but she's not convinced this is a responsibility of the Basque government.

Figure 5: **Qualitative comparison on GoodNews.** We cut the articles to fit the figure size. The entity names from image information are highlighted in light purple (PERSON tag) and light blue (ORG tag) colors. We can see that the named entities in captions also appear in the human-written and ENGINE-generated articles. In contrast, the GPT2-generated and GROVER-generated articles do not contain correct entity names corresponding to the image.

| Method | Human | RoBERTa |
|---|---|---|
| GROVER-Mega | 72.8% | 90% |
| GPT2-1.5B (Finetuned) | 69.6% | 84% |
| ENGINE-XL | 67.6% | 84% |

Table 3: **Detection of generated articles.** Left column reports the performance of AMT workers at correctly identifying an article as human or machine generated. Right column uses OpenAI's text detector based on RoBERTa (Liu et al., 2019) to perform the same task.

| Method | 124M | 355M | $\geq$ 1.5B |
|---|---|---|---|
| GPT2 | 26.1 | 19.1 | 14.8 |
| GPT-Neo | 24.9 | **13.1** | 11.5 |
| GPT-J-6B | - | - | **9.0** |
| GPT2 (Finetuned) | 33.8 | 25.2 | 25.9 |
| ENGINE (NE) | **20.7** | 15.4 | 16.3 |

Table 4: **Zero-shot article generation.** We compare the perplexity of ENGINE to GPT-2 (Radford et al., 2019), GPT-Neo (Gao et al., 2020), and GPT-J (Wang and Komatsuzaki, 2021) on WikiText (Stephen et al., 2017). Both GPT2 and ENGINE are finetuned on Visual-News (Liu et al., 2021) with a maximum article length of 1024. Though the data distribution between Wikipedia and news is different, our proposed entity-aware method still improves the performance over several baselines.

sistent with our annotation experiment, we compare the human-written article with three machine-generated articles. From the results, we can see that ENGINE-XL can effectively produce articles with the named entities learned from image information. In contrast, finetuned GPT2-1.5B and GROVER-Mega failed to generate correct named entities in articles. For example, both ENGINE-generated article and the human-written article mentioned "Hean Louis Iratzoki" and "Alki", which are appeared in the caption. In contrast, articles generated by GPT2 or GROVER are discussing some other entities such as "Chris Burch" and "Ms. Madrid."

## 4.3 Article Retrieval

Table 5 compares the Recall@K retrieval scores of ENGINE with the state-of-the-art baselines on the

| Method | Image-to-article | | | | | | Article-to-image | | | | | |
| | GoodNews | | | VisualNews | | | GoodNews | | | VisualNews | | |
| | R@1 | R@5 | R@10 | R@1 | R@5 | R@10 | R@1 | R@5 | R@10 | R@1 | R@5 | R@10 |
|---|---|---|---|---|---|---|---|---|---|---|---|---|
| CLIP-B/16 | 19.4 | 42.7 | 52.9 | 30.6 | 53.5 | 64.6 | 32.5 | 56.8 | 64.4 | 43.1 | 69.5 | 76.5 |
| CLIP-L/14 | 32.5 | 56.0 | 65.3 | 47.3 | 72.1 | 78.9 | 46.6 | 68.3 | 75.0 | 55.0 | 77.3 | **84.3** |
| BLIP | 19.8 | 38.0 | 48.8 | 22.0 | 43.6 | 52.5 | 15.9 | 30.8 | 37.9 | 15.1 | 33.3 | 42.7 |
| EVA01-CLIP-G/14 | 49.6 | 70.3 | 77.1 | 56.9 | 78.9 | 84.7 | 47.6 | 68.3 | 75.2 | 54.9 | 76.5 | 82.5 |
| EVA02-CLIP-L/14 | 50.7 | 73.1 | 79.4 | 57.9 | 78.6 | 85.2 | 49.3 | 69.6 | 76.8 | 54.8 | 77.3 | 82.7 |
| ENGINE (ours) | **53.8** | **73.5** | **79.6** | **61.9** | **82.0** | **86.7** | **51.5** | **72.0** | **77.8** | **56.3** | **78.3** | 83.9 |

Table 5: **Retrieval results.** We compare the Recall@K of ENGINE to CLIP (Radford et al., 2021), BLIP (Li et al., 2022), and EVA-CLIP (Sun et al., 2023) on the article-to-image and image-to-article retrieval tasks on GoodNews and VisualNews. For articles with multiple embedded images, we only use the first embedded image. See Section 4.3 for discussion.

test splits of GoodNews and VisualNews. Following Tan et al. (2022), we randomly select 1500 article-image pairs from each dataset for evaluation. In the image-to-article retrieval task, we observe that compare to EVA02-CLIP-L/14 (Sun et al., 2023), ENGINE boost the Recall@1 scores from 50.7 to 53.8 on GoodNews and from 57.9 to 61.9 to VisualNews. For article-to-image retrieval, ENGINE achieves the highest performance with Recall@1 scores of 51.5 and 56.3 on GoodNews and VisualNews, respectively. The retrieval results validate the importance of named entities within articles and the effectiveness of our proposed method.

## 5   Discussion

In our paper, we mainly investigate modeling machine-generated articles, which can be used directly for generation while also can providing strong language features to support applications like article retrieval. However, actors can also use the same technology to generate articles for misinformation by modifying information of specific fields to realize two potential purposes: monetization (ad revenue through clicks) or propaganda (communicating targeted information) (Zellers et al., 2020). Thus, the development of a better article generator can not only help humans write high-quality articles but also potentially help train a more powerful discriminator. Table 3 reports the performance of using human judgements or OpenAI's RoBERTa-based machine generated text detector (Liu et al., 2019). When comparing the results of the RoBERTa detector for GPT2-1.5B and ENGINE-XL, we find that only 25% of the articles that were predicted as human written came from the same generation prompts. Thus, the two methods can provide differ-

ent views of the same prompt, which can provide additional information for training an even more powerful machine generated text detector. We note that our contributions are largely architecture agnostic, so they could also be used in RNN-based generators, which may provide a larger distribution shift in the generated articles that may fool a discriminator trained only on Transformer outputs.

## 6   Conclusion

In this paper, we proposed ENGINE, an entity-aware article generation and retrieval method that explicitly incorporates and models named entities in language models. Concretely, ENGINE extracts named entities from both metadata and embedded images in articles, providing a more comprehensive source of entity information. In addition, we introduce an entity-aware mechanism to help ENGINE recognize and predict named entities more effectively and accurately. ENGINE outperforms current popular language models in quantitative and qualitative experiments on GoodNews, VisualNews, and WikiText. For example, ENGINE outperforms GPT-J by roughly 2.5 perplexity points using only a quarter parameters of GPT-J and boost the performance of EVA02-CLIP by 3-4 Recall@1 accuracy in article retrieval experiments. The noticeable improvements demonstrate that ENGINE can generate and retrieve articles more accurately and efficiently by effectively leveraging named entities.

**Acknowledgements** This material is based upon work supported, in part, by DARPA under agreement number HR00112020054. Any opinions, findings, and conclusions or recommendations expressed in this material are those of the author(s) and do not necessarily reflect the views of the supporting agencies.

## Limitations

We discuss limitations and potential improvements of our work in this section. First, though our method can effectively predict the correct entity names in articles, their corresponding entity categories might be mistakenly predicted. For example, in Figure 5, the brand name "Alki" is recognized as a city name by ENGINE. Therefore, a more accurate entity-aware mechanism could be developed in future work. Second, the image information can be further explored. In this paper, we mainly investigate the captions and named entities of news images. However, other information such as the locations of images within articles may also prove useful for article generation. In addition, our current methods detect named entities from images considering each entity independently using a text-image matching framework. However, since the relationships between entities also affect the probability that entities appear in images, the incorporation of entity relationships can also be considered to further improve the entity detection module.

## Ethics Statement

ENGINE is a model for article generation and retrieval. It can either help automated journalism or defending against machine-generation articles. However, there is no perfect system which can generate 100% accurate articles. Therefore, it is critical for practitioners to check the fact mentioned in articles and avoid the misinformation brought by failure generation cases. Additionally, someone could use our approach to generate misinformation. However, ENGINE applies the network structure that is same to GPT2, which means the discriminators trained for GPT2 articles (*e.g.*, RoBERTa detector (Liu et al., 2019)) are also effective to discriminate ENGINE-generated articles. Our paper helps to highlight the need for building tools like the RoBERTa detector.

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

# A  Additional Experiment Results

## A.1  Article Generation

Additional qualitative results on article generation are provided in Figure 6, 7, 8, 9, 10, 11, 12 as the supplement of our main paper, demonstrating that ENGINE is able to generate articles given different news sources.

## A.2  Article Quality Annotation

Table 6 reports human accuracy in identifying articles as machine or human-generated to supplement the main paper. We see that ENGINE-XL is able to generate more realistic-looking news articles on Goodnews, consistent with the conclusion in our main paper.

| Human-based detector | |
|---|---|
| GROVER-Mega | 76.4% |
| GPT2-1.5B (Finetuned) | 73.6% |
| ENGINE-XL | **70.4%** |

Table 6: The performance of AMT workers at correctly identifying an article as human or machine generated on GoodNews.

## A.3  Ablation Study on Top-k Named Entities

We provide the ablation study on top-k named entities in Table 7. We see that the model achieves the best performance when $k$ is set to 15. When $k$ is greater than 10, the improvement is limited.

| top-k named entities | 5 | 10 | 15 | 20 |
|---|---|---|---|---|
| PPL $\downarrow$ | | 15.5 | 14.8 | 14.5 | 14.6 |

Table 7: Ablation study on the number of named entities detected by CLIP (GoodNews).

## A.4  Recall of CLIP-detected Named Entities

We use Oracle NE as ground truth labels to evaluate the retrieval results by CLIP model. In Table 8, we observe that approximately 30% named entities from Oracle NE have been retrieved by CLIP. In contrast, recall of Cap on GoodNews and VisualNews are 22.57% and 7.41% respectively. The gap between GoodNews and VisualNews is likely because captions in GoodNews are often much longer than captions in VisualNews. The retrieval results validate that ClipNE contains more relevant named entities compared to named entities extracted solely based on captions.

| Named Entities | GoodNews | VisualNews |
|---|---|---|
| Cap | 22.57 | 7.41 |
| ClipNE | 29.84 | 31.19 |

Table 8: Recall of Cap, ClipNE on GoodNews and VisualNews. Cap represents named entities that appear in captions, ClipNE represent named entities detected by CLIP model.

## A.5 Ablation Study on the Canonical Order

The ablation study of varying inference order is shown in Table 9. From the Table, we see that canonical orders which are not consistent with the training order result in greater PPL of the language model. The model achieves the best performance when the inference order is aligned with the training order.

| canonical order | GoodNews PPL↓ |
|---|---|
| date-domain-title-summary | 18.2 |
| title-date-domain-summary | 19.2 |
| summary-date-domain-title | 20.5 |
| domain-date-title-summary | 17.3 |

Table 9: Ablation study on the canonical order during inference (GoodNews).

## A.6 Ablation Study on Article Retrieval

We ablate the two modules of our method on article retrieval in Table 10. Text-only denotes that we directly use the original articles as the input. NE denotes named entity extraction, and EA denotes the entity-aware mechanism. From the table, we observe that both named entity extraction and the entity-aware mechanism can boost the article retrieval performance.

| Method | Image-to-article | | | Article-to-image | | |
|---|---|---|---|---|---|---|
| | R@1 | R@5 | R@10 | R@1 | R@5 | R@10 |
| Text-only | 57.9 | 78.6 | 85.2 | 54.8 | 77.3 | 82.7 |
| NE | 60.0 | 80.3 | 85.9 | 56.1 | 78.2 | 83.7 |
| NE-EA | **61.9** | **82.0** | **86.7** | **56.3** | **78.3** | **83.9** |

Table 10: Ablation study on article retrieval (Visual-News). Both named entity extraction and the entity-aware mechanism can improve the article retrieval performance on VisualNews.

## A.7 Decoding Strategy for Article Generation

Likelihood-maximization decoding strategies like greedy search or beam search work well in close-ended generation such as image captions, machine translation, or summarization. However, these methods suffer from the repetitive text problem in open-ended generations like dialog or story generation (Hashimoto et al., 2019; Holtzman et al., 2019). Sampling methods (Fan et al., 2018; Holtzman et al., 2019) are therefore proposed to introduce more randomness and surprise to text generation. In our work, we adopt the top-p sampling (nucleus sampling) method (Holtzman et al., 2019) as our decoding strategy.

## A.8 Article Quality Annotation Templates

Following (Tan et al., 2020), annotators are asked to indicate a reason for whether the articles are human-written or machine-manipulated. We provide a view of the AMT worker interface in Figure 13.

Title: Daniel Agger leaves Liverpool to return to former club Brondby
Domain: www.theguardian.com
Date: 08-30-2014

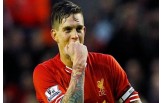
Daniel Agger has rejoined Brondby for an undisclosed fee.

**Human-written News Article**

Liverpool have announced the sale of the defender Daniel Agger to the Danish club Brondby for an undisclosed fee. The 29-year-old Denmark international joined the Anfield club from Brondby in 2006 and went on to make more than 200 appearances for the club but had fallen down the pecking order under Brendan Rodgers.

"Liverpool have been such a big part of my life and my family's lives for so long, that leaving is extremely difficult," Agger said. "The opportunity arose for me to return home to Brondby and at this stage of my career it felt like the right decision for me. "I wouldn't leave here to go anywhere else and that has been proven by my actions in recent seasons - I have turned down many offers to move to other Premier League and European clubs. "I would like to thank the Liverpool supporters for the incredible backing they have given me in my time here and the warmth and generosity of spirit displayed to my family. "This has been our home and somewhere we have been very happy and settled. However, more recently I have started to feel the effects of playing regularly in a league that is as physically demanding as the Premier League is, and I would not want to stay if I did not think that I was able to meet that challenge week in, week out."

**ENGINE-generated News Article**

The Liverpool defender Daniel Agger has left Anfield to return to his previous club Brondby. The Danish international midfielder was reported to have agreed personal terms with the club. The 29-year-old left Liverpool in 2006 following their relegation from the Premier League and has featured for the national team in Denmark's recent seasons.

The move to Brondby was said to be a " cash - plus " deal. " We're very pleased Daniel Agger has chosen to continue his development at Brondby, " read a statement on the club website. " He's a young man who has shown great potential in the Premier League, and we look forward to working with him again. " Brondby, who are in the Danish Superliga, have a squad of more than 200 players. The Liverpool manager Brendan Rodgers said : " As a footballer you dream of playing in the Premier League and I'm very pleased that Daniel Agger has chosen to come back to Brondby. " He's a good defender and I'm sure he will work very hard to prove to the club that he is a good player. " We like to think of him as a very important player for his nationality. But he has a great character. You might think he's a bit of a character but you 'll see he's as strong a character as any of the other players here. " Agger said : " I am extremely proud to play for Brondby. " I will always consider it as one of the happiest days of my career so far. It has been a great and very memorable time, and the supporters have been fantastic. My future lies with the club, and hopefully, it will be here for a long time. " I also want to thank the Liverpool fans for their support. Without them I wouldn't have been able to carry on fighting for this shirt, so thank you! "

Figure 6: Example of human-written and ENGINE-generated news articles from *Guardian*.

**Title: State of Britain′s wildlife ′increasingly fragile′**
Domain: www.bbc.com
Date: 12-08-2015

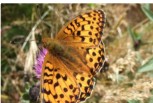 Dark green fritillary butterfly

Human-written News Article

Britain′s wildlife is in an increasingly fragile state, with animals carrying out vital jobs for farmers being lost more rapidly than others, say scientists.Insects that pollinate crops or fight pests are at risk of disappearing, putting food production in jeopardy, according to the team.The research brings together millions of wildlife records spanning 40 years.It suggests conservation efforts should focus on certain areas, they add.Dr Tom Oliver of the University of Reading, who led the research, said it was the biggest and most comprehensive report ever assembled for any country in the world."By standardising records from an army of amateur biologists across the country, we have amassed an impressive array of data, giving us our most complete picture yet of the state of Britain′s wildlife," he told BBC News. "The picture that emerges is of an increasingly fragile system, particularly in species that do vital jobs for humans. "Unless efforts are made to reverse some of these declines, we face a future where we will be less confident that we can effectively grow our food."Benefits from nature.

The researchers looked at records of the changing fortunes of more than 4,000 types of plant and animal living in England, Wales and Scotland between 1970 and 2009.Species such as bees, dragonflies, grasshoppers, ladybirds, plants and mosses were put into groups based on the role they play in nature.Groups providing pollination and pest control benefits had undergone declines while those involved in functions such as decay or mopping up carbon emissions were more stable.Plants and animals regarded as of cultural importance to humans, such as birds, butterflies and hedgehogs, also fared badly.Prof James Bullock, of the Centre for Ecology and Hydrology in Oxfordshire, a co-researcher on the study, said conservation efforts should focus on the areas where there was strong evidence of declining fortunes."Conservation actions, such as wildlife friendly farming, can avoid the loss of biodiversity and the resulting erosion of the pollination, pest control and other benefits we derive from nature," he said.And Dr Oliver said there were a number of measures that individuals could take to help to improve the future for wildlife in the British Isles.They include: Making gardens wildlife friendly Purchasing food grown in a sustainable way to wildlife Helping to collect data on wildlife. The study is published in the journal Nature Communications.

ENGINE-generated News Article

The state of Britain′s wildlife has become increasingly fragile due to loss of habitat and changes in the climate, a study has suggested. Mass bird and butterfly losses have increased along with climate change, scientists found, which means many species are not as well equipped to cope. And while changes in food distribution, such as floods, may cause short - term problems, they can make long - term problems worse. Drought and flood events could put millions of birds and insects at risk, the scientists concluded. Monitoring of more than 4,000 species of birds, butterflies and moths across England, Wales and Scotland for 40 years concluded that the UK′s biodiversity was still very good - only smaller than it was in 1975.But the biological communities in which they live were now less diverse than when these earlier surveys took place, which meant their ability to recover was less certain, they added. "We have lost several species of already endangered species, " explained lead researcher Dr Tom Oliver, a naturalist at the University of Reading. "And we are moving into many of these habitats that are becoming drier and thus less suitable for them. "The results of the study were " far more alarming " than expected. "The situation looks pretty dire, " said Dr James Bullock from the Centre for Ecology and Hydrology at Oxfordshire, who was not involved with the research. He told BBC News that Britain′s wildlife was already extremely stressed. " Our natural environment is suffering from climate change, which is melting the ice sheets and expanding the oceans, and from human - generated pollution, so we are going to have a much more polluted environment in the future. " And the species we are losing are particularly important because when you lose species, you do so in numbers, they cease to exist in numbers. " The loss of a single species in Britain alone has the potential to affect millions of other organisms. "The impact of the findings is likely to be felt most in many communities across the British Isles. "We have lost a large number of species, " Dr Oliver added. " It′s going to have a big impact on the ecology of Britain. " He explained that the risk of long - term damage was unlikely to be taken lightly : " We are more concerned about short - term impacts. "The researchers believe that much work needs to be done to understand how, in time, this situation might change. Research in Nature Communications suggests that climate change means there has been a sharp decline in the number of butterfly species spotted in northern England - down by a third between 1970 and 2009.Species such as the large blue have struggled for many generations as the cold weather has made it too cold to overwinter in the UK.Follow Helen on Twitter.

Figure 7: Example of human-written and ENGINE-generated news articles from *BBC*.

Title: Robberies plague Tenleytown area over the past few weeks
Domain: www.washingtonpost.com
Date: 05-23-2016

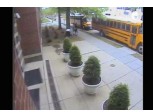 Video shows suspects robbing attacking Georgetown Day student

**Human-written News Article**

A student at Georgetown Day School was robbed last month in the same area of the District where a student at another school was stabbed during a robbery attempt, and near where two other robberies also occurred last month.The student at Georgetown Day was robbed about 1:30 p.m. April 27 in the 4200 block of Davenport Street , near the private school. Three robbers approached, and began beating the victim, police said, although they did not identify him as a student.They went through his pants pocket and took his glasses, wallet and phone before fleeing on foot, police said.A spokeswoman for Georgetown Day issued a statement saying that a student there was robbed along 42nd Street in Tenleytown on April 27, and the school provided immediate medical and security support.The statement said the school has held an assembly devoted to neighborhood safety training and it has adjusted the patrols of school security officers.That robbery occurred two days after an April 25 incident in which authorities said a student at Woodrow Wilson High School was stabbed in the leg during an attempted robbery near the school.In the Wilson incident, the robber was described as a man in his 30s. No description was provided of the robbers on Davenport Street NW. Police released video Monday of three people described as "persons of interest."The video showed the victim and the robbers approaching each other on a sidewalk, with school buses lined up at a curb. Suddenly the student is knocked down and one or two of the robbers appear to go through his pockets.But the video does not appear to make possible a detailed description of the robbers.The stabbing comes nearly a week after two robberies, one of them involving a student victim, were reported on streets near Wilson and near Georgetown Day. Both occurred within about 15 minutes of each other April 19. In each instance, three people, one of whom who was armed with a knife, approached female victims and demanded their cellphones and debit cards. A juvenile male has been arrested in those cases, police said.

**ENGINE-generated News Article**

A group that robbed Georgetown Day School of valuables including cash last month and a purse while it was locked up in a dormitory has also robbed a spate of Georgetown Day students, police said. The initial thefts occurred April 27 and April 27. Two men entered the school, stealing the wallets and purses of two female students, police said. A total of four men were released from custody. The school said the total was $ 170.The last robbery occurred about 1:30 p.m. April 27 near the store at 4200 Davenport Street. Three male suspects fled the school on foot, and a local resident engaged in an effort to help the robbers, police said. It took about 15 minutes before the local man was able to locate them. The three defendants - one in his 20s, one in his late 20s and one in his late teens - fled along Davenport Street NW and into a residential neighborhood, police said. That resident was identified as a friend of the robbers and assisted in helping apprehend them. The robbery at Georgetown Day came two days after a robbery at Woodrow Wilson High School in Tenleytown. That April 25 robbery took place near 42nd Street and Davenport Street NW, a major thoroughfare in Tenleytown. According to police, two men in their 20s robbed a woman in her 30s. The robbers fled with a purse containing cash that was in a pink leather wallet. On Monday, the Wilson robbery went unsolved for nearly a week. Staff at the school said no incidents had been reported. "We do have a staff member watching the campus. You go through the normal procedures of setting up alarms, " said a Tenleytown police spokeswoman. " We were n't able to get that information from the yet. The person you are talking to is someone within security. You have to keep an eye on him. "Correction : This story has been updated to reflect that the Davenport Street robbery occurred April 19, not April 27.

Figure 8: Example of human-written and ENGINE-generated news articles from *Washington Post*.

**Title: Courts again rule against streaming upstart Aereo**
Domain: www.usatoday.com
Date: 08-21-2014

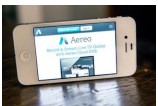 In this photo illustration Aereocom a Web service that provides television shows online is shown on an iPhone 4S on April 22 2014 in New York City

**Human-written News Article**

A federal court will not hear Aereo's argument that the streaming media company should be allowed to operate similarly to a cable company.The Second Circuit Court of Appeals ruled Thursday that Aereo will have to take its argument to the district court, according to court documents first reported by The Washington Post.This decision continues broadcasters' winning streak against the upstart service. Launched two years ago, Aereo delivered over-the-air channels via online streaming to computers and tablets for about $8 a month and had brought the service to 11 cities.The Supreme Court ruled two months ago that Aereo was violating copyright law by not paying the networks for transmitting their signals.After that, the streaming media company voluntarily shut down and told the U.S. District Court that it wanted to be treated as the equivalent to a cable TV system and apply for a compulsory license to stream live TV broadcasts.Subsequently, the Copyright Office's general counsel said that the office did not think that Aereo would qualify, but fell short of denying Aereo's filing because the issue remained alive in the courts.Follow Mike Snider on Twitter: @MikeSnider

**ENGINE-generated News Article**

The high court delivered yet another blow to Aereo in a pair of unanimous court opinions Thursday declaring the popular streaming service illegally alters the contents of customers'cable subscriptions and is guilty of violating copyright law. The Second Circuit Court of Appeals, which first ruled against Aereo two years ago, urged the justices to reject its claim that it is simply a device for consumers to access live TV without paying for it. Aereo lets consumers use a digital antenna to receive live broadcasts online and streams those broadcasts as they are being transmitted over the Internet, just like cable TV.Through those transmissions the consumer also becomes a subscriber to the over - the - air service. But even if they keep Aereo's service on, only a portion of the amount paid for cable service remains. For example, for $ 16.95 per month Aereo customers receive nearly all the TV shows they request, regardless of what shows they watch. For about $8 each the consumer loses the signals they might already have over the Internet. Thus the company is violating copyright law by redistributing the service users already pay for without paying for the service they are receiving from cable operators. "The Court finds that Aereo's business model violates this section of the Copyright Act, " 11 judges wrote in the majority opinion. The Washington Post, which first reported on the decision, called it " a blow to digital media and to consumers. ""This unanimous decision means a lot, " Aereo chief executive Mike Snider said in a statement. " Consumers no longer have to pay to receive the traditional way they watch TV. They now have the option to pay only for a service that delivers those channels, all for free. "Aereo has petitioned the court to change the district court's ruling to the full appeals court, which is likely to do so. The Supreme Court upheld the same ruling from two months ago. Aereo said it will appeal to the U.S. District Court for the District of Columbia. Experts tell the Post that its previous case shows that the Copyright Office's determination to give consumers access to streaming services without paying retailed companies is likely to be reversed by the appeals court. Follow @jswartz on Twitter

Figure 9: Example of human-written and ENGINE-generated news articles from *USA Today*.

**Title: James F. Holland, Trailblazing Cancer Researcher, Dies at 92**
Domain: www.nytimes.com
Date: 03-27-2018

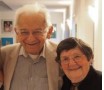 Dr. Holland with his wife, Dr. Jimmie Holland, who pioneered the field of psycho-oncology. While she treated the emotional distress of cancer patients, he focused on discovering universal remedies. She died in December.

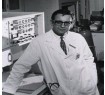 Dr. James F. Holland in an undated photograph. In the 1950s, he and his colleagues were considered research cowboys by some colleagues, he said.

**Human-written News Article**

James F. Holland, a founding father of chemotherapy who helped pioneer a lifesaving drug treatment for pediatric leukemia patients, died on Thursday at his home in Scarsdale, N.Y. He was 92.

The cause was respiratory failure, his son Steven said.

Dr. Holland and his medical collaborators in the 1950s were scorned by some colleagues as research "cowboys," as he put it, because they audaciously tackled what was considered an incurable disease and tried to treat it with several chemicals simultaneously, instead of sequentially.

When their research began, fewer than one in three children who were found to have acute lymphoblastic leukemia lived as long as a year. Today, as many as nine in 10 survive the disease, in which the bone marrow makes too many immature white blood cells.

Another clinical trial that Dr. Holland directed in the 1970s, to contain acute myeloid leukemia, a somewhat similar malfunction in blood cell production, resulted in what became the standard treatment.

**ENGINE-generated News Article**

James F. Holland, an innovative oncologist whose scientific insights made him, as he once said, a " Google " for cancer research, died on Thursday at his home in Scarsdale, N.Y. He was 92.

The cause was a widespread myeloid leukemia subtype, a cancer of the blood cells in bone marrow that occurs in fewer than one in three patients.

Mr. Holland was one of the few oncologists to propose, in the 1950s, the chemical treatment known as stereosporin -- hence the name. He was also among the first to suggest ( though not actually carry out ) what became known as precancer therapy : turning over nonmedical patients to their regular doctors, while supplementing their therapies with medications to kill cancer cells. The mere suggestion of precancer therapy, it seemed to some, would have rendered it toxic.

At the time, though, Mr. Holland thought it should not. He and his colleagues had been working for as long as a year on drugs to prevent acute lymphoblastic leukemia in patients with advanced disease, in which cells on the surface of the blood can proliferate out of control after being overexposed by chemotherapy. Doctors usually did this only after the patient's health deteriorated. Today, many hospitals will not perform precancer therapy, either.

But in the 1970s, he and his colleagues were regarded as " research cowboys " by their colleagues. Known for treating as many as nine of 10 cancer cases, they had been derided even by their own specialty, nuclear medicine, for treating these patients when " the science does not justify it, " Dr. Steven Holland, his brother and his only sibling, said.

Figure 10: Example of human-written and ENGINE-generated news articles from *New York Times*.

## Robert Boulter

Robert Boulter is an English film, television and theatre actor. He had a guest @-@ starring role on the television series The Bill in 2000. This was followed by a starring role in the play Herons written by Simon Stephens, which was performed in 2001 at the Royal Court Theatre. He had a guest role in the television series Judge John Deed in 2002. In 2004 Boulter landed a role as \" Craig \" in the episode \" Teddy's Story \" of the television series The Long Firm ; he starred alongside actors Mark Strong and Derek Jacobi. He was cast in the 2005 theatre productions of the Philip Ridley play Mercury Fur, which was performed at the Drum Theatre in Plymouth and the Menier Chocolate Factory in London. He was directed by John Tiffany and starred alongside Ben Whishaw, Shane Zaza, Harry Kent, Fraser Ayres, Sophie Stanton and Dominic Hall. \n\n In 2006, Boulter starred alongside Whishaw in the play Citizenship written by Mark Ravenhill. He appeared on a 2006 episode of the television series, Doctors, followed by a role in the 2007 theatre production of How to Curse directed by Josie Rourke. How to Curse was performed at Bush Theatre in the London Borough of Hammersmith and Fulham. Boulter starred in two films in 2008, Daylight Robbery by filmmaker Paris Leonti, and Donkey Punch directed by Olly Blackburn. In May 2008, Boulter made a guest appearance on a two @-@ part episode arc of the television series Waking the Dead, followed by an appearance on the television series Survivors in November 2008. He had a recurring role in ten episodes of the television series Casualty in 2010, as \" Kieron Fletcher \". Boulter starred in the 2011 film Mercenaries directed by Paris Leonti. \n\n In 2000 Boulter had a guest @-@ starring role on the television series The Bill ; he portrayed \" Scott Parry \" in the episode, \" In Safe Hands \". Boulter starred as \" Scott \" in the play Herons written by Simon Stephens, which was performed in 2001 at the Royal Court Theatre. A review of Boulter's performance in The Independent on Sunday described him as \" horribly menacing \" in the role, and he received critical reviews in The Herald, and Evening Standard. He appeared in the television series Judge John Deed in 2002 as \" Addem Armitage \" in the episode \" Political Expediency \", and had a role as a different character \" Toby Steele \" on The Bill. \n\n He had a recurring role in 2003 on two episodes of The Bill, as character \" Connor Price \". In 2004 Boulter landed a role as \" Craig \" in the episode \" Teddy's Story \" of the television series The Long Firm ; he starred alongside actors Mark Strong and Derek Jacobi. Boulter starred as \" Darren \", in the 2005 theatre productions of the Philip Ridley play Mercury Fur.

Robert Boulter is an English film, television and theatre actor. He had a guest @-@ starring role on the television series The Bill in 2000. This was followed by a starring role in the play Herons . He has also starred on TV as a teacher and actor. In 2004 Boulter starred in the stage play Waste in Plymouth. He made his film debut at the Festival theatre in London in the 2005 John Sabatalain adaption of the John Grant - written novel, Intimate Behaviour. In 2006 he won a Samuel Beckett award for Best Performance in a Television Play for the film version of his play Teddy's Story. In 2007 Boulter played the lead role in the adaptation of Patrick Burden's book, East of Eden. In 2008 he was involved with Dominic Hall and John Deed and Craig Mark Strong, who were co - writer of another film, the comedy thriller, Derek Jacobi And The Mercenaries, and the TV adaptation of the Hot Fuzz movies. In May 2008 he made his film acting debut in the film Fever. Since then he has been sought out for appearing in films and TV series including Mercury Fur, which he appeared in as John Tiffany ; Hammersmith, where he played Ben Whishaw ; and London's Bush Theatre, where he appeared alongside Shane Zaza in Man of Economic Mass. In November 2008 he returned to the stage, in West End production of This Side of Paradise and in the West End production of Until Goes. He is currently appearing at the Shakespeare theatre at Fulham with a revival of Gods and Heroes. Robert Boulter biography Being called a fusion of two performers has never been embarrassing in Boulter's theatre career. In taking the offbeat, gruff character of Teddy that readers may instantly recognise as Harry Kent he defied expectations ; it was an unexpected and welcome return from a genre that had seen him move towards cheerfulness, his name becoming subtly attached to his image. He has also made the perfect transition from film to stage, propelling himself between brutal, seriously unpleasant characters on screen and harsh, adoring protagonists with dignity and aplomb in real life. Boulter lives with his family in Burn, Devon. His son Fraser Ayres is a member of the team behind the website jeffersonsworld.com, and his daughter Sophie Stanton has been a character actor in the theatre. Boulter starred alongside Whishaw in the 2010 film Donkey Punch. The other main roles in the film were Mark Ravenhill, who plays a former officer in an augmented forces unit, and Josie Rourke as his. In the play the lead character, \" the demon of myth \" is C, who lives in the arctic wasteland and is transformed when he is savagely hunted by a face - eating gnactus. Boulter played the first half of the story, or lion, while the other half was The Soldier, a mercenary, \" the curse of mythology \".

Figure 11: Example of human-written and ENGINE-generated Wikipedia articles (from WikiText dataset (Stephen et al., 2017)). We generate articles conditioned on both the title **Robert Boulter**, and the first 50 tokens highlighted by light yellow. The articles are cut to fit the figure size.

## 1933 Treasure Coast hurricane

**Human-written Wikipedia**

The 1933 Treasure Coast hurricane was the second @-@ most intense tropical cyclone to strike the United States during the active 1933 Atlantic hurricane season. The eleventh tropical storm, fifth hurricane, and the third major hurricane of the season, it formed east @-@ northeast of the Leeward Islands on August 31. The tropical storm moved rapidly west @-@ northwestward, steadily intensifying to a hurricane. It acquired peak winds of 140 miles per hour ( 225 km / h ) and passed over portions of the Bahamas on September 3, including Eleuthera and Harbour Island, causing severe damage to crops, buildings, and infrastructure. Winds over 100 mph ( 161 km / h ) affected many islands in its path, especially those that encountered its center, and many wharves were ruined. \n\n Subsequently, it weakened and made landfall at Jupiter, Florida, early on September 4 with winds of 125 mph ( 201 km / h ). The hurricane moved across the state, passing near Tampa before moving into Georgia and dissipating. In Florida, the strong winds of the cyclone blew buildings off their foundations, and numerous trees were prostrated in citrus groves. The Treasure Coast region received the most extensive destruction, and Stuart, Jupiter, and Fort Pierce were heavily damaged. Inland, the cyclone weakened rapidly but produced prodigious amounts of rain, causing a dam to collapse near Tampa. The storm caused $ 3 million in damage ( 1933 USD ) after damaging or destroying 6 @,@ 848 homes. \n\n Unusually, the storm hit Florida less than 24 hours before another major hurricane bearing 125 @-@ mile @-@ per @-@ hour ( 201 km / h ) winds struck South Texas ; never have two major cyclones hit the United States in such close succession. \n\n The origins of the hurricane were from a tropical wave that possibly spawned a tropical depression on August 27, although there was minimal data over the next few days as it tracked to the west @-@ northwest. On August 31, a nearby ship reported gale force winds, which indicated that a tropical storm had developed to the east @-@ northeast of the Lesser Antilles. Based on continuity, it is estimated the storm attained hurricane status later that day. Moving quickly to the west @-@ northwest, the storm passed north of the Lesser Antilles and Puerto Rico. Early on September 2, a ship called the Gulfwing reported a barometric pressure of 978 mbar ( 28 @.@ 88 inHg ), which confirmed that the storm attained hurricane status. After passing north of the Turks and Caicos islands, the hurricane struck Eleuthera and Harbour Island in the Bahamas on September 3, the latter at 1100 UTC. A station on the latter island reported a pressure of 27 @.@ 90 inHg ( 945 mb ) during the 30 minute passage of the eye.

**ENGINE-generated Wikipedia**

The 1933 Treasure Coast hurricane was the second @-@ most intense tropical cyclone to strike the United State during the active 1933 Atlantic hurricane season. The eleventh tropical storm, fifth hurricane. A Category 4 cyclone with winds of 100 mph generated winds of 161 km per hour and a water temperature of -49 degrees. The storm killed 6 people, flooded an area of 848 sq km, and caused $ 3 million in damage. Although it missed the Florida coast, the system swept across the rest of the United States less than 24 hours. The heat from the system caused widespread fires and continued to bring heavy rain and wind throughout the south and central regions. In south Florida alone, the fires killed two men and destroyed 98 homes in the Lesser Antilles and southwest Florida. The Turks and Caicos islands were particularly badly hit because of a storm surge that produced high seas and flooding, especially along the coast from Tampa, Georgia, to Puerto Rico. The Turks and Caicos islands experienced over 125 mph winds, winds of 125 @-@ mile per hour, and only 1 cm to 2 inches of rain in the 30 minute interval starting around 1400 UTC. The Turks and Caicos islands also experienced an over 140 mph wind shear at 5875 kp / m. In south Georgia, home to the Simpson and Penally counties, the worst of the damage was felt in the Savannah area, where 800 + homes were damaged or destroyed and well over 1000 had to be demolished. The most intense tropical storm to strike the United States during the 1933 Atlantic hurricane season occurred when Tropical Depression 6 formed over the Bahamas during the winter of 1913. The storm resulted in catastrophic loss of life and destruction. As of 24 km west of the Bahamas, the lowest pressure was 3.21 mb, and the storm had maximum sustained winds of 200 km/h, with gusts to over 200 km/h. In south Florida, the intense storm produced winds of 100 mph over south - western southwest Florida and a south - western coast generally between the edges of the Low Range and the extreme southeast of the storms. In western Cuba, the storms produced winds up to what the Weather Bureau described as \" about 65 mph \" in some spots, and all tropical storms at Category 4, 5 and 7, respectively, were downgraded to a tropical storm or depression. Texas though was hit by Hurricane Norbert in early October 1913. At the time, the Weather Bureau reported that the storm produced winds of 800 - 900 km/h, and a storm tide of 848 m. The storm passed over northeast Texas between midnight and dawn, and was the first major tropical cyclone to impact the United States during the winter of 1913. Norbert resulted in 848 lives being lost and $ 98 m in damages, amongst other catastrophic damage, as well as numerous fatalities.

Figure 12: Example of human-written and ENGINE-generated Wikipedia articles about **1993 Treasure Coast hurricane**.

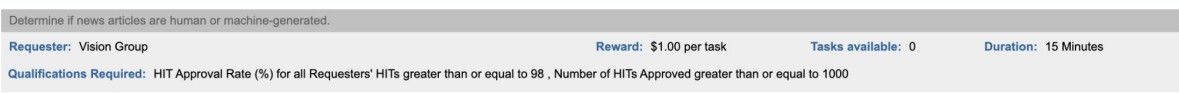

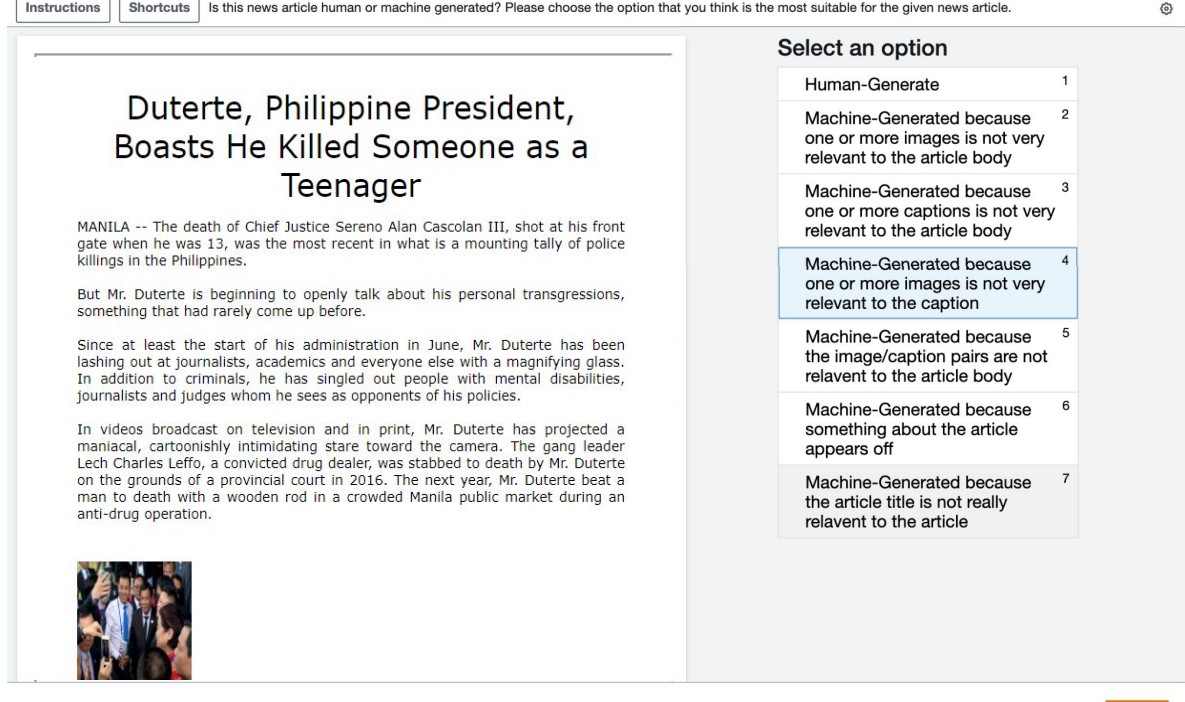

Figure 13: The interface used by AMT workers in our article quality annotation experiment.