# OpenReview forum: "Show, Write, and Retrieve: Entity-aware Article Generation and Retrieval"
_EMNLP/2023/Conference — EMNLP 2023 Findings_

### Official Review · Reviewer_KMxz · 2023-08-01

**Soundness:** 3

**Excitement:**

4: Strong: This paper deepens the understanding of some phenomenon or lowers the barriers to an existing research direction.

**Paper Topic And Main Contributions:**

This paper studies article comprehension task with applications including article generation and image-to-article retrieval, and proposes an ENtity-aware article GeneratIoN and rEtrieval approach, ENGINE , to explicitly incorporate and model named entities in articles. Experiments on GoodNews and VisualNews demonstrate that ENGINE can boost the performance of article generation and retrieval. The authors also perform comprehensive experiments on human evaluation and machine discrimination, validating that ENGINE produces more realistic articles compared to prior work.

**Reasons To Accept:**

This paper explicitly incorporates and models named entities in articles to boost the performance of article generation and retrieval, which is reasonable and effective in both theory and practice. The authors have also carried on various experiments to demonstrate that ENGINE can produce high-quality and hard-to-detect articles, which can potentially contribute additional training data for the development of more powerful machine-generated text detectors.

**Reasons To Reject:**

1.In CLIP-based NER, the authors select top k entities directly to represent the image, which remains to be explored. The balance between the accuracy and randomness of the entity should be studied more deeply.
2.The training process of ENGINE isn’t clearly illustrated. Please add loss function, lear ing rate and other training strategies to explain how the model is finetuned.
3.The ablation study of ENGINE is incomplete. It is suggested that the authors should add Cap-CapNE-ClipNE-EA、Cap-CapNE-ClipNE-NE-EA according to the overview model of ENGINE. Also, the authors should give more detailed analysis to the ablation results.
4.The experiments on article retrieval are inadequate. The authors can add ablation study of ENGINE on article retrieval.

**Reproducibility:**

4: Could mostly reproduce the results, but there may be some variation because of sample variance or minor variations in their interpretation of the protocol or method.

**Reviewer Confidence:**

4: Quite sure. I tried to check the important points carefully. It's unlikely, though conceivable, that I missed something that should affect my ratings.

---

> ### Author Rebuttal · Authors · 2023-08-29
>
> We thank the reviewer for their valuable comments, we appreciate their time and will use their suggestions to improve our paper.
>
> > In CLIP-based NER, the authors select top k entities directly to represent the image, which remains to be explored. The balance between the accuracy and randomness of the entity should be studied more deeply.
>
> We provide the experiments of different $k$ values in Appendix A.3, Table 7. We see that when $k$ is greater than 10, the improvement of the language model is limited. Therefore, we set the value of $k$ to 10 to balance the accuracy and randomness of the named entities. Below we recreate Table 7 related to this experiment for your convenience.
>
> **Ablation study on the number of named entities detected by CLIP (GoodNews).**
> |   top-k entities   | 5 | 10 | 15 | 20 |
> |----------- |:--------:| :--------:| :--------:| :--------:|
> | PPL |  15.5  | 14.8 | 14.5 | 14.6 |
>
>
> > The training process of ENGINE isn’t clearly illustrated. Please add loss function, lear ing rate and other training strategies to explain how the model is finetuned.
>
> We implemented our models mainly based on Pytorch [3] and Transformer [4] libraries.
> The maximum sequence length of language models is set to 1024. For ENGINE-Base and ENGINE-Medium, we used a batch size of 8 and a maximum learning rate of $10^{-4}$. For ENGINE-XL, we used a batch size of 4 to fit the GPU memory. Correspondingly, the maximum learning rate is set to $2^{0.5}\times10^{-4}$. We trained our models around 3 epochs with 0.06 epoch for linear warm-up on both datasets. We parallelized ENGINE-XL on 4 NVIDIA RTX-A6000s and ENGINE-Medium on 2 NVIDIA RTX-A6000s. Consistent to other GPT models, we apply negative log-likelihood as our loss function. **We will also release our code to ensure reproducibility in the camera ready.**
>
>
> > The ablation study of ENGINE is incomplete. It is suggested that the authors should add Cap-CapNE-ClipNE-EA、Cap-CapNE-ClipNE-NE-EA according to the overview model of ENGINE. Also, the authors should give more detailed analysis to the ablation results.
>
> Following the reviewer's suggestions, we provide additional comparisons here as the supplement of our ablation study in Figure 3.
>
> **Ablation study on VisualNews and GoodNews.**
> |   Method      | VisualNews PPL | GoodNews PPL |
> |-------------------------------|:----------:|-----------:|
> | Text-only | 18.3 | 17.3 |
> | Cap | 17.7 | 16.6 |
> | Cap-EA | 17.1 | 15.9 |
> | Cap-CapNE-EA | 17.1 | 16.0 |
> | Cap-ClipNE-EA | 16.1 | 14.8 |
> | **Cap-CapNE-ClipNE-EA** | 16.0 | 14.8 |
> | **Cap-CapNE-ClipNE-NE-EA** | 13.2 | 12.3 |
> | Cap-NE-EA | 13.1 | 12.0 |
>
> We bolded the newly added ablation experiments to better differentiate them from existing ones. From the table, we observe that combining entities using CapNE and ClipNE (*Cap-CapNE-ClipNE-EA*) did not provide an improvement over *Cap-ClipNE-EA*. This is because CapNE is extracted from *Cap*. When *Cap* is available as input to language models, *CapNE* does not introduce any new information to language models. Similarly, we also observe that *Cap-CapNE-ClipNE-NE-EA* does not outperform *Cap-NE-EA* for the same reason.
>
> > The experiments on article retrieval are inadequate. The authors can add ablation study of ENGINE on article retrieval.
>
> We ablate the two modules of our method in the following Table.
>
> **Ablation study on Article Retrieval. From left to right (R@1-R@10): Image-to-article on GoodNews, Image-to-article on VisualNews, Article-to-image on GoodNews, Article-to-image on VisualNews**
>
> |   Method      | R@1 | R@5 | R@10 | R@1 | R@5 | R@10 | R@1 | R@5 | R@10 | R@1 | R@5 | R@10 |
> |-------------------------------|:----------:|-----------:|:----------:|-----------:|:----------:|-----------:|:----------:|-----------:|:----------:|-----------:|:----------:|-----------:|
> | Text-only | 50.7 | 73.1 | 79.4 | 57.9 | 78.6 | 85.2 | 49.3 | 69.6 | 76.8 | 54.8 | 77.3 | 82.7 |
> | NE | **54.1** | **74.6** | **80.8** | 60.0 | 80.3 | 85.9 | 51.3 | **72.1** | 77.4 | 56.1 | 78.2 | 83.7 |
> | NE-EA | 53.8 | 73.5 | 79.6 | **61.9** | **82.0** | **86.7** | **51.5** | 72.0 | **77.8** | **56.3** | **78.3** | **83.9** |
>
>
>
> Text-only denotes that we directly use the original articles as the input. NE denotes named entity extraction, and EA denotes the entity-aware mechanism. From the table, we observe that both named entity extraction and the entity-aware mechanism can boost the article retrieval performance. In most experiment settings, we find performance gains introduced by the entity-aware mechanism. In the image-to-article experiments on GoodNews, we did not observe improvements brought by the entity-aware mechanism, but we argue this is simply due to the noise introduced by entity category recognition on GoodNews, which may affect the performance of this experiment.

---

### Official Review · Reviewer_U6FG · 2023-08-03

**Soundness:** 2

**Excitement:**

2: Mediocre: This paper makes marginal contributions (vs non-contemporaneous work), so I would rather not see it in the conference.

**Paper Topic And Main Contributions:**

This paper proposes an entity-aware article generation and retrieval approach to explicitly incorporate named entities into language models. Because the applications of real-world events may reference many named entities that are hard to be recognized by language models. The experiments on three public datasets demonstrate the effectiveness of the proposed model.

**Reasons To Accept:**

The authors propose to extract the named entities from both metadata and embedded images associated with articles, and utilize an entity-aware mechanism that enhances the model’s ability to recognize and predict entity types. The experimental results demonstrate the usefulness of the proposed framework.

**Reasons To Reject:**

The authors propose the CLIP-based Visual NER framework to extract the entities. And I want to know whether the parameters of the framework contain the CLIP-based Visual NER module. Because the framework also needs to utilize Visual NER module to extract the entities during the inference time.

**Reproducibility:**

4: Could mostly reproduce the results, but there may be some variation because of sample variance or minor variations in their interpretation of the protocol or method.

**Reviewer Confidence:**

4: Quite sure. I tried to check the important points carefully. It's unlikely, though conceivable, that I missed something that should affect my ratings.

**Typos Grammar Style And Presentation Improvements:**

Please keep the formula reference format consistent, such as “Eq.” and “Equation”.

---

> ### Author Rebuttal · Authors · 2023-08-28
>
> We thank the reviewer for the comments, we appreciate their time and will use their suggestions to improve our paper. Since the reviewer lowered the scores before we post our responses, we sincerely hope that the following responses can address the concerns.
>
> > The authors propose the CLIP-based Visual NER framework to extract the entities. And I want to know whether the parameters of the framework contain the CLIP-based Visual NER module. Because the framework also needs to utilize Visual NER module to extract the entities during the inference time.
>
> As our goal is to compare language models, we did not consider the effect of modules not related to language generation as they can simply be swapped out with varying methods or entirely excluded, as we do when using user provided named entities.  For example, we provide an example of user provided named entities in our oracle entities, whereas Visual NER compares those extracted using CLIP, but this could also come from another means.  However, as the main point of comparison is the model performing language generation, this is where we make our primary comparison.  This also follows the views of how to count model parameters in prior work. For example, VisualGLM-6B [1] introduces BLIP-2 [2] as the image encoder to enable image comprehension, and they did not include the parameters of BLIP-2 when naming their model.  However, even if we include the parameters of the CLIP model we used, which is 151M, the total parameters of ENGINE and CLIP (1.5B+151M) is still much smaller than GPT-J (6B) and GPT-Neo (2.7B), **demonstrating that including CLIP’s parameters in the total parameter count of our model would not change any of our conclusions.**
>
> [1] Du, Zhengxiao, et al. "GLM: General Language Model Pretraining with Autoregressive Blank Infilling", ACL (2022).
> [2] Li, Junnan, et al. "Blip-2: Bootstrapping language-image pre-training with frozen image encoders and large language models." arXiv (2023).
>
>
> >Please keep the formula reference format consistent, such as “Eq.” and “Equation”.
>
> We thank the reviewer for this feedback. We shall use a more consistent formula reference format in our camera ready.

---

### Official Review · Reviewer_5dtH · 2023-08-03

**Soundness:** 4

**Excitement:**

3: Ambivalent: It has merits (e.g., it reports state-of-the-art results, the idea is nice), but there are key weaknesses (e.g., it describes incremental work), and it can significantly benefit from another round of revision. However, I won't object to accepting it if my co-reviewers champion it.

**Missing References:**

1. Do not understand this representation in Abstract: "4-5 perplexity gain". Can it be more explicit like a single number.

2. For the image, can't we get the meta-data information from the news articles?

3. Using Visual NER can be noisy with CLIP, as it is only passed with Visual Information and not multimodal.

4. Since an external tool (SpaCy) is used for NER, how do we know the precision and accuracy of annotation? Also, the dataset itself do not have any ground truth to support it.

5. In section-3.2, it is not clear how CLIP predicts the similarity between the article images and the candidate entities. As, named entities are just phrases with maximum length of 2 or 3 words. However, clip training was performed on image-text pairs that are sentences.

6. Does changing the canonical order impact the results?

7. Ablation study lack a study on how jointly modeling entity names and their categories is boosting the performance of article generation and retrieval. Aforementioned statement was stated as the benefits of proposed approach.

8. Not clear about the human study for this use-case on asking humans to distinguish machine-generated articles from human-written articles. It make sense in case of fake news. It will be good to add more motivation.


**Paper Topic And Main Contributions:**

In this paper, authors propose a an entity-aware article generation and retrieval approach (ENGINE) for incorporation of named entities into language models. Experiments are conducted on three different datasets mainly GoodNews, VisualNews and a zero-shot article generation on WikiText showcase the effectiveness of the proposed approach against several other strong language model baselines on metrics such as Perplexity and Recall@K.

**Reasons To Accept:**

1. Adds the entity category name beside the recognized named entity for the proposed approach.
2. Extensive comparison of results with other language models and closely related tasks.

**Reasons To Reject:**

1. Human conducted study do not correlate well with the use-case of this paper.
2. Unclear about the precision of named entities acquired from the articles as the dataset lack ground truth information.
3. Assumption of CLIP trained with a large dataset contains named entities. However, this has not be quantified.

**Reproducibility:**

1: Could not reproduce the results here no matter how hard they tried.

**Reviewer Confidence:**

4: Quite sure. I tried to check the important points carefully. It's unlikely, though conceivable, that I missed something that should affect my ratings.

---

> ### Author Rebuttal · Authors · 2023-08-28
>
> We thank the reviewer for their valuable comments, we appreciate their time and will use their suggestions to improve our paper.
>
> > Human conducted studies do not correlate well with the use-case of this paper.
> > Not clear about the human study for this use-case on asking humans to distinguish machine-generated articles from human-written articles. It makes sense in the case of fake news. It will be good to add more motivation.
>
> In the article generation task, the ability of humans to distinguish between the two sources is an important measurement, since generative language models are trained to replicate the distribution of human-written content. This is consistent with the arguments made in prior work, such as GPT-3 [1] and GROVER [2], who performed similar experiments to evaluate the quality of generated articles (*e.g.*, Table 7.3 in GPT-3 [1], Figure 4 in GROVER [2]).
>
> [1] Brown, Tom, et al. "Language models are few-shot learners." NeurIPS (2020).
> [2] Zellers, Rowan, et al. "Defending against neural fake news." NeurIPS (2019).
>
> > Unclear about the precision of named entities acquired from the articles as the dataset lacks ground truth information.
> > Since an external tool (SpaCy) is used for NER, how do we know the precision and accuracy of annotation? Also, the dataset itself does not have any ground truth to support it.
>
> We agree that spaCy [3] is not a perfect predictor of named entities from articles. However, by leveraging the detected named entities, we still observe a perplexity improvement of our language models. The improvement further validates that our proposed model is effective, even in condition that the named entities are not the ground truth labels and may contain noise. In summary, using detected named entities with noise, our method already outperforms the baselines. Our experiments with oracle named entities also show that we can expect greater improvements as NER methods become more accurate or the list of entities is provided as input.
>
> [3] Honnibal, M., Montani, I., Van Landeghem, S., & Boyd, A. (2020). spaCy: Industrial-strength Natural Language Processing in Python.
>
> > Assumption of CLIP trained with a large dataset contains named entities. However, this has not been quantified.
>
> The training data of CLIP [4] is a dataset of 400 million image-text pairs collected from the Internet. The text data in the Internet inherently contains a great number of named entities. Since OpenAI does not release their training data of CLIP, we can check another image-text pairs dataset which are also collected from the Internet, LAION-400M [5]. Upon simply reviewing the HuggingFace tutorial of LAION-400M (search for laion-400M on the HuggingFace datasets page), we see that the captions of LAION-400M contain a lot of named entities.
>
> To validate this assumption quantitatively, we provide retrieval experiments in Appendix A.4, Table 8.  We use Oracle NE as ground truth labels to evaluate the retrieval results by CLIP model. From the Table, we observe that approximately 30% named entities from Oracle NE have been retrieved by CLIP. Compared to named entities that appear in captions, ClipNE improves the recall by 7 points on GoodNews and 24 points on VisualNews. Below we recreate Table 8 related to this experiment for your convenience.
>
> **Recall of Cap, ClipNE on GoodNews and VisualNews.**
> |  Named Entities  | GoodNews | VisualNews |
> |----------- |:--------:| :--------:|
> | Cap |  22.57 | 7.41 |
> | ClipNE | 29.84 | 31.19 |
>
> [4] Radford, Alec, et al. "Learning transferable visual models from natural language supervision." International conference on machine learning. PMLR, 2021.
> [5] Schuhmann, Christoph, et al. "Laion-400m: Open dataset of clip-filtered 400 million image-text pairs." arXiv (2021).
>
> > Do not understand this representation in Abstract: "4-5 perplexity gain". Can it be more explicit like a single number?
>
> The improvement varies with the scale of our model. For example, ENGINE-Base improves around 5.3 perplexity points compared to finetuned GPT-124M, while ENGINE-XL improves around 3.9 perplexity points compared to finetuned GPT-1.5B. Therefore, this defines a range of a 4-5 perplexity improvement in the Abstract. We shall clarify this in the camera ready.
>
> > For the image, can't we get the meta-data information from the news articles?
>
> In the article generation task, the objective of language models is to produce articles based on human-written prompts. These articles are primarily considered as the model's output. Thus, we do not assume that the news articles are available as part of the input and, therefore, the meta-data *cannot* be obtained from the news articles during inference.
>
>
> > Using Visual NER can be noisy with CLIP, as it is only passed with Visual Information and not multimodal.
>
> We would like to begin by noting that our primary contribution is introducing a new entity-aware mechanism that takes advantage of NERs more effectively than the entity-aware mechanisms proposed by prior work (reported in Table 2b of our paper).
>
> However, when demonstrating what situations are useful, we did evaluate the effectiveness of using automatically extracted NERs, such as the Visual NERs.  Nearly all detector outputs can introduce some noise, as can many datasets due to label noise. However, the key question is whether these provide a benefit that is greater than that of the noise they introduce.  As demonstrated in our experiments, using Visual NER boosts performance, which means that they are effective enough to be beneficial. Specifically, our ablation study in Figure 3 shows that the CLIP-detected named entities improve our model by around 1 perplexity points on both GoodNews and VisualNews datasets (GoodNews:15.9 vs. 14.8, VisualNews: 17.1 vs. 16.1).
>
> Finally, since we use an off-the-shelf image-text model, this suggests that as these models (or others) become more accurate at identifying NERs, our proposed entity aware mechanism will simply become more effective in these settings.
>
>
> > In section-3.2, it is not clear how CLIP predicts the similarity between the article images and the candidate entities. As, named entities are just phrases with maximum length of 2 or 3 words. However, clip training was performed on image-text pairs that are sentences.
>
> In the CLIP paper, the authors performed zero-shot experiments on ImageNet using sentences like "a photo of <object category>," demonstrating one way to adapt CLIP to a classification task with a short (often single word) label. We also note that in the official code repository of CLIP, the authors also use short phrases such as "a diagram", "a dog“, and "a cat" as the language examples.  This suggests that CLIP should work well for very short phrases as well as longer prompts created using templates.  For our task we found these different methods of using CLIP for short phrases made negligible differences in practice.  Specifically, we followed the same experimental setup in Appendix A.4, Table 8, where we evaluate how many of the oracle named entities are extracted either using CLIP or from the image captions.  We found that over 500 randomly selected samples, the recall of using the template “a photo of <named entity>” was 30.254, whereas using the named entity alone obtained a recall of 30.253, demonstrating that either approach works equally well (in our case, we used the named entity directly).
>
> > Does changing the canonical order impact the results?
>
> We observe that our model performs at its best when the inference order aligns with the order used during training. For example, during training, we set the order as "domain-date-title-summary". The ablation study of varying inference order is shown in the following table.
>
> **Ablation study on the canonical order during inference (GoodNews).**
> |   canonical order   | GoodNews PPL |
> |----------- |:--------:|
> | date-domain-title-summary | 18.2 |
> | title-date-domain-summary | 19.2 |
> | summary-date-domain-title | 20.5 |
> | domain-date-title-summary | **17.3** |
>
> From the Table, we see that canonical orders which are not consistent with the training order result in greater PPL of the language model. The model achieves the best performance when the inference order is aligned with the training order.
>
>
> > Ablation study lack a study on how jointly modeling entity names and their categories is boosting the performance of article generation and retrieval. Aforementioned statement was stated as the benefits of proposed approach.
>
> We performed ablation studies on jointly modeling entity names and their categories in Figure 3 (*Cap* vs. *Cap-EA*). We see that our entity-aware mechanism improves the perplexity by approximately 0.6 points (16.6 vs. 15.9 on GoodNews and 17.7 vs. 17.1 on VisualNews). To supplement this comparison, we also finetune GPT-2 with our entity-aware mechanism (*Text-EA*). We find that compared to *Text-only*, *Text-EA* achieves 0.4 and 0.5 perplexity improvements on GoodNews and VisualNews, respectively. The additional comparison further validates the effectiveness of our proposed entity–aware mechanism.
>
>
> > Implementation details for reproducibility:
>
> We implemented our models mainly based on Pytorch [3] and Transformer [4] libraries.
> The maximum sequence length of language models is set to 1024. For ENGINE-Base and ENGINE-Medium, we used a batch size of 8 and a maximum learning rate of $10^{-4}$. For ENGINE-XL, we used a batch size of 4 to fit the GPU memory. Correspondingly, the maximum learning rate is set to $2^{0.5}\times10^{-4}$. We trained our models around 3 epochs with 0.06 epoch for linear warm-up on both datasets. We parallelized ENGINE-XL on 4 NVIDIA RTX-A6000s and ENGINE-Medium on 2 NVIDIA RTX-A6000s. **We will also release our code to ensure reproducibility in the camera ready**.

---

### Meta-Review · Area_Chair_DZzy · 2023-09-20

**Recommendation:** 4

**Metareview:**

Proposes an approach for article generation that is enhanced with two key components: a NER module and an entity-aware component. Experiments on 3 public datasets show significant gain on perplexity and recall@k.

PLUS:

- extensive comparison with LMs on closely related tasks.
- ENGINE can potentially contribute valuable training data for machine generated text detectors.

MINUS:

- human study does not correlate with the main use case.
- precision of NEs obtained from articles?
- Missing details about loss function, learning rate and training strategies for ENGINE.
- incomplete ablation study.

---

### Decision · Program_Chairs · 2023-10-07

**Decision:**

Accept-Findings

**Comment:**

Proposes an approach for article generation that is enhanced with two key components: a NER module and an entity-aware component. Experiments on 3 public datasets show significant gain on perplexity and recall@k.

PLUS:

- extensive comparison with LMs on closely related tasks.
- ENGINE can potentially contribute valuable training data for machine generated text detectors.

MINUS:

- human study does not correlate with the main use case.
- precision of NEs obtained from articles?
- Missing details about loss function, learning rate and training strategies for ENGINE.
- incomplete ablation study.